# Research on the Measurement and Path of Urban Agglomeration Growth Effect †

Jing Han *, Ming Gao * and Yawen Sun

School of Economics and Resource Management, Beijing Normal University, Beijing 100875, China; zbsyw9423@sina.com
* Correspondence: nkhanj@bnu.edu.cn (J.H.); joyreal0607@163.com (M.G.)
† Based on the Empirical Research of 14 National Urban Agglomerations in China.

**Abstract:** This paper employed dynamic generalized method of moment methods to measure the growth effect of 202 prefecture-level cities covered by 14 national urban agglomerations in China from 2007 to 2016. Based on this, this paper further explored the main factors affecting the growth of urban agglomeration and the path to achieving sustainable growth from the aspects of system, technology, structure, and influencing factors, and used the dynamic panel data (DPD) model and threshold panel data to empirically test the growth effect of urban agglomerations. The empirical results showed the following. (1) From the perspective of influencing factors, the improvement of technology and the increase in technology expenditure had a good growth effect on urban agglomeration, and this growth effect became more and more significant as the economic development level within the urban agglomeration narrowed; moreover, the increase of the agglomeration degree could alleviate the negative externality caused by the expansion of the urban scale and produce the dispersion effect to relieve the pressure of urban agglomeration. (2) From the results of the growth effect of urban agglomerations, the growth effect of multi-core urban agglomerations was more significant than that of single-core and dual-core urban agglomerations, and technology, agglomeration degree, foreign direct investment and human capital all significantly promoted the growth of urban agglomerations. Compared with trans-provincial urban agglomerations, provincial urban agglomerations have less resistance due to administrative jurisdiction, and the growth effect was obvious. (3) From the perspective of regional differences, the growth momentum of urban agglomerations in the eastern region was significantly stronger than that in the central and western regions, and the growth effect of agglomeration degree, technology, and human capital on urban agglomeration were all stronger than that in the central and western regions. Considering that the spatial distance between the edge cities and the central cities of the urban agglomeration will have an important impact on the overall growth of the urban agglomeration, this paper then used the panel threshold method to deeply discuss the influence mechanism and path dependence of the agglomeration degree on the growth of urban agglomerations. The results showed that within a certain spatial scale, a higher agglomeration degree of an urban agglomeration creates a stronger radiation effect of the core city and more obvious growth momentum of the urban agglomeration. In the future development of urban agglomerations, it is necessary to clarify the functions of the core city, vigorously develop new technologies, strengthen the construction of the core city as well as maximize its radiation and driving effect on the surrounding cities. Meanwhile, the government should improve transportation, increase the construction of urban expressways and railways, strengthen the connection between cities, strengthen regional integration and cooperation, and give play to the role of human capital in promoting growth to achieve the stable and continuous growth of urban agglomerations.

**Keywords:** urban agglomeration; growth effect; radiation effect; agglomeration degree

## 1. Introduction

Urban agglomeration is a collection of cities with central cities as the core that radiate to the surrounding areas. China introduced the 11th Five-Year Plan in 2006 to promote urban agglomeration as the main form of urbanization. In 2014, the National New Urbanization Plan identified four national urban agglomerations, nine regional urban agglomerations, and six local urban agglomerations, which, from the perspective of spatial structure, further emphasized the development of urban agglomerations with high agglomeration efficiency, a strong radiation effect, and complementary advantages to have an important impact on national economic growth and regional coordinated development. Subsequently, in 2015 and 2016, the Government Work Report and the 13th Five-Year Plan respectively emphasized the importance of urban agglomeration construction for regional coordinated development and the future development of the country, and proposed to further establish and improve the coordination mechanism for urban agglomeration development, strengthen inter-regional urban linkages, and achieve the efficient development of urban agglomerations. On 18 November 2018, the Communist Party of China and the state council issued vital opinions on establishing a new, more effective mechanism for regional coordinated development that represents the proposal of the national major regional strategy promoted by urban agglomerations such as the Beijing–Tianjin–Hebei urban agglomeration, the Yangtze River Delta urban agglomeration, the Guangdong–Hong Kong–Macao Greater Bay Area (former Pearl River Delta urban agglomeration and Hong Kong and Macao), the Chengdu–Chongqing urban agglomeration, the urban agglomeration in the middle reaches of the Yangtze River, the Central Henan urban agglomeration, and the Guanzhong Plain urban agglomeration, which has led to the formal establishment of a new model in which the central city leads the development of urban agglomerations and urban agglomerations drive regional development.

At present, China's urban agglomerations cover nearly 22% of the total land area, 49% of the total population, 79% of the total economic output, 70% of fixed assets investment, 85% of the tertiary industry added value, and 98% of foreign investment. The major urban agglomerations (the Yangtze River Delta, the Pearl River Delta, and the Beijing–Tianjin–Hebei region) as well as the middle reaches of the Yangtze River and the Chengdu–Chongqing urban agglomerations have gathered 40% of the population with an area of 11%, creating 55% of the GDP. Urban agglomeration has gradually established the "main body" form in the process of urbanization development, which has an important influence on regional economic growth. Therefore, it is of great practical significance to measure the growth effect of urban agglomeration from the perspective of future urbanization development and the sustainable growth of the regional economy.

The continuous agglomeration of cities promotes the integration of cities of different sizes in a certain region and promotes the development of urban agglomerations through the continuous flow of population and other factors. However, the growth of urban agglomeration is not driven by a single factor, and city scale, human capital, investment of technological research and development, spatial distance and industrial structure, etc. exert influences on economic growth from various aspects [1–3]. With the rise of the third technological revolution and the advancement of innovative national strategies, the impact of the development and application of new technologies on urban economic structure and regional integration has gradually deepened, and economic growth has spread across geographically limited areas while promoting knowledge spillovers and the improvement of the technical level. However, it cannot be ignored that the economic development gap between cities is still significant and persistent [4–7]. Due to the differences in the size of China's cities, the problem of unbalanced development between regions has become more prominent, which has caused a polarization trend of rapid expansion of megacities and the relative shrinkage of small and medium-sized cities and small towns. The difference in economic development between regions is obvious, in particular, the difference between North and South has far exceeded the difference between East and West. Therefore, the differences of the growth effect of urban agglomerations and the internal development mechanism of different regions need to be further studied in combination with the geographical location and economic development foundation of different cities.

The good growth effect of urban agglomerations also needs the promotion of central cities. The theory of central land defines the formation and establishment of a central city in urban agglomeration. The differentiation of development between cities is in line with the development law of objective things. With spatial agglomerations and expansions of the city scale, urban agglomerations are in need of one or several places with good development, large markets, and advantageous locations as their regional centers. At the same time, central cities with a high density of economy and mature degree of capital accumulation could transfer in terms of factors, industries, technologies, and talent to the surrounding cities. By virtue of their superior geographical locations and obvious industrial advantages, central cities will have "siphon effects" on surrounding cities, which could promote regional coordinated development and produce a good growth effect on urban agglomeration [8–10]. However, the effect of agglomeration on urban economic growth is not linear. In some countries, when urban agglomerations develop to certain stages, some hierarchical systems will be generated to make agglomerations in the marginal areas have positive impacts on urban economic growth, while the agglomerations in the core areas have a hindrance to growth and there are different levels of monopolistic competition, which lead to a significant difference in economic growth between urban agglomerations [11]. Due to the different development levels of central cities, excessive population agglomeration and rising transportation costs will bring about a series of "big city diseases", causing a significant diffusion force of central cities with a developed economy and high economic density to surrounding cities and a disconnection between the development of the central cities of the underdeveloped urban agglomeration and other cities, which is not conducive to regional coordinated development [12–14]. The development of the central cities to the urban agglomerations is mainly reflected in leading and driving the development of the surrounding cities of the urban agglomeration, realizing the transfer of elements, and spilling out through knowledge transformation and innovation as well as constantly improving the core foundation configuration and establishing the innovation system to achieve collaborative growth. Therefore, to establish a core city, it is not only necessary to consider the size and development degree of the core city, but also its spatial distance from its surrounding cities and its own development potential as well as whether it has an open, inclusive, and diversified urban culture [15–17].

So how does urban agglomeration produce good growth effects?

Good growth effects are reflected not only in the highly developed economy and advanced industrial structure, but also in the sustainability of the economic growth path. Due to the difference in geographic distance between the surrounding cities and the core cities and the size of the cities, there will be a significant gap in the agglomeration degree of urban agglomerations. In theory, the higher the agglomeration degree of urban agglomerations, the more the cities they contain, and the higher the level of economic development, the stronger the growth momentum of the urban agglomerations, and the more significant the growth effect. However, the excessive agglomeration of cities creates a "crowding effect", which is not conducive to the sustainable growth of urban agglomerations in the future. Is there a reasonable threshold for the agglomeration degree of urban agglomerations?

Compared with previous studies, this paper has the following three marginal contributions. First, it breaks through the perspective of a traditional comparative study of urban agglomerations, makes an in-depth analysis of the development mode of south and north urban agglomerations, and finds out the differentiated development path of south and north urban agglomerations. Second, it not only discusses the urban agglomerations with a different number of core cities by sample, but also discusses the economic growth effect of urban agglomerations under different administrative divisions to determine the future development trend of urban agglomerations. Third, it integrates the agglomeration degree and economic growth of urban agglomerations into a unified framework for analysis, and further analyzes the influence mechanism of the reasonable threshold of agglomeration degree on urban agglomeration growth.

The rest of this paper is arranged as follows. Section 2 is the theoretical model and research hypothesis. Section 3 is the empirical test of the growth effect of urban agglomerations in China.

Section 4 is the robust test. Section 5 further studies the inner action mechanism of the agglomeration degree on urban agglomeration growth. Section 6 presents our conclusions and suggestions of this paper.

## 2. Theoretical Model and Research Hypothesis

Based on the endogenous growth model which includes that technology is constructed, we used an adjusted Solow–Swan growth model to measure the growth effect [18,19]. Then, we introduced human capital, the agglomeration degree, and other factors using constant returns to scale and Hicks' neutrality to set up the basic model. The improved Cobb–Douglas production function form is:

$$Y = A_t K_t^\alpha Q_t^\beta L_t^{1-\alpha-\beta} \tag{1}$$

In Equation (1), A is technology, which generally comes down to the contribution of the output level to the combination of technological improvement or institutional change; K is the capital stock; Q is the human capital stock; L is the total labor force; and $\alpha$, $\beta$ are the parameters of the production function. $0 < \alpha < 1$, $0 < \beta < 1$, and $\alpha + \beta = 1$ when the scale returns are constant, t is the period. Divide both sides of Equation (1) by L to get:

$$\frac{Y_t}{L_t} = A_t K_t^\alpha Q_t^\beta L_t^{-\alpha-\beta} \tag{2}$$

Then,

$$y_t = A k_t^\alpha q_t^\beta L_t^{-\alpha-\beta} \tag{3}$$

where $y_t$ is the output per capita in the year t; $k_t = K_t/L_t$; and $q_t = Q_t/L_t$.

Assuming that the form of production function of material capital is certain, the depreciation rate is fixed, and the growth rate of population is exogenous, that is, $\dot{L}_t/L_t = n$. Therefore, the following dynamic equation can be obtained:

$$\dot{k}_t = s_k f(k_t) - (n+\delta)k_t = s_k A_t k_t^\alpha q_t^\beta L_t^{\alpha+\beta} - (n+\delta)k_t \tag{4}$$

$$\dot{q}_t = s_q f(q_t) - (n+\delta)q_t = s_q A_t k_t^\alpha q_t^\beta L_t^{\alpha+\beta} - (n+\delta)q_t \tag{5}$$

Assuming that technology is constant, the stable state level of material capital per capita and human capital per capita obtained from Equations (4) and (5) are:

$$k^* = \left(\frac{s_k^{1-\beta} s_q^\beta}{n+\delta} A\right)^{\frac{1}{-\alpha-\beta}} l_t \tag{6}$$

$$q^* = \left(\frac{s_k^{1-a} s_q^a}{n+\delta} A\right)^{\frac{1}{-\alpha-\beta}} l_t \tag{7}$$

So, the output per capita at the stable state level is:

$$y_t^* = \frac{1}{(n+\delta)} A^{\frac{1}{-\alpha-\beta}} s_k^{\frac{\alpha}{-\alpha-\beta}} s_q^{\frac{\beta}{-\alpha-\beta}} l_t \tag{8}$$

Assuming $s_k = s_q = s$ [20], and if $\mu = -\alpha - \beta$, $-\mu = \alpha + \beta$ take the logarithm of Equation (8), we will obtain:

$$\ln y_t^* = \frac{1}{\mu} \ln A - \mu \ln s + \ln l_t \tag{9}$$

According to the treatment of Mankiw et al. (1992) to the linear model under stable state, we can obtain:

$$\frac{d\ln y}{dt} \approx \frac{d\ln(y/y^*)}{dt} \equiv H = -\mu[\alpha\ln(k/k^*) + \beta\ln(q/q^*)]$$
$$= -\mu(\ln y - \ln y^*) \tag{10}$$

Combining Equations (9) and (10), we can obtain:

$$H = (1-\mu)\ln A - \mu\ln y + \mu^2\ln s - \mu(1-\mu)\ln l \tag{11}$$

In order to explain the impact of agglomeration on output growth, this paper assumes that urban agglomeration growth is not only affected by factor inputs, but also by the degree of urban agglomeration. In order to fully consider the spatial and population size factors of cities, referring to the relevant processing method of cluster degree coefficient by PortNov, B.A and Schwartz (2009), this paper introduced the ratio of spatial isolation degree (IS) and marginal degree (IR) of cities to measure the cluster degree of the city (for more specific treatment refer to the data selection), assuming:

$$H = (1-\mu)\ln A - \mu\ln y + \mu^2\ln s - \mu(1-\mu)\ln l + e^{\theta IC_t + \lambda_t} \tag{12}$$

where A represents technological level; H is the economic growth rate; e is the random error term and other factors affecting output growth; the degree of the cluster(IC) is the agglomeration degree of an urban agglomeration in the year t; θ is the parameter; and λ is the random error term.

Then, after introducing the error term and other factors that may affect the growth, the following equation can be obtained:

$$H = \alpha_0 + \alpha_1\ln A + \alpha_2\ln y + \alpha_3\ln s + \alpha_4\ln l + \theta IC + \varepsilon \tag{13}$$

Among the explanatory variables, we were most concerned about the impact of technological level, GDP per capita, and clustering (IC) on growth. By observing the statistical significances and the signs of the estimated coefficients in the regression results, we could judge whether the growth effect of urban agglomeration was positive or negative. On this basis, the following hypotheses are proposed in this paper:

**Hypothesis 1.** *The higher the technical level of the urban agglomeration, the higher the economic growth rate of the urban agglomeration and the more significant the growth effect.*

Technology has a significant impact on both urban and urban agglomeration economies. The improvement of technology can optimize resource allocation in a larger space, improve resource utilization, further realize the flow of factors, and promote the growth of urban agglomeration through regional coordinated development.

**Hypothesis 2.** *The smaller the urban economic development gradient within the urban agglomeration, the more obvious the economic growth effect of the urban agglomeration, which means that the smaller the difference of y between cities of urban agglomerations, the greater the growth effect of urban agglomeration significance.*

With the improvement of the urban economic development stage, each city relies on its own resource endowment and location advantages to carry out professional development. However, if the level of urban development is very different, it may be difficult to achieve effective complementarity, and the greater the siphon effect of the core city.

**Hypothesis 3.** *Within a certain spatial range, before reaching the critical point of agglomeration, the higher the agglomeration degree of urban agglomeration, the stronger the growth effect.*

Agglomeration plays a significant role in promoting the development of the area. Areas with significant agglomeration have more employment opportunities, more reasonable internal division of labor, and better public facilities and welfare. However, after the agglomeration reaches a certain level, the competition between cities will become more intense, and the "crowding effect" and "urban disease" will appear. Therefore, the agglomeration of cities is not the higher the better.

## 3. Empirical Test

### (1) Sample and Data Selection

This paper selected the statistical data of the prefecture-level city level in 2008–2017 to measure the growth effect of 14 national urban agglomerations (The 14 urban agglomerations are: the Beijing–Tianjin–Hebei urban agglomeration, the Yangtze River Delta urban agglomeration, the Pearl River Delta urban agglomeration, the Central Henan urban agglomeration, the urban agglomeration in the middle reaches of the Yangtze River, the Guanzhong Plain urban agglomeration, the Chengdu–Chongqing urban agglomeration, the Harbin–Changchun urban agglomeration, the urban agglomeration of the middle and south Liaoning, the Beibu Gulf urban agglomeration, the Hohhot–Baotou–Erdos–Yulin urban agglomeration, the Lanzhou–Xining urban agglomeration, the Shandong Peninsula urban agglomeration, and the urban agglomeration on the west coast of the Taiwan Straits) that have been confirmed and are waiting for approval, and each year includes 202 prefecture-level city samples. In this paper, some cities with serious missing data were excluded as well as Hong Kong and Macao data that have not been published in the statistical yearbook, and the missing data of some cities in individual years were filled linearly. On one hand, it can avoid the occurrence of special values in a certain year and cause errors in the overall population and economic level of the city. On the other hand, the multicollinearity of the variables can be avoided to further study the dynamic adjustment process.

In this paper, the data used for controlling variables of each prefecture-level city such as regional GDP, GDP per capita, population of municipal districts, proportion of three industries in GDP, foreign direct investment, and proportion of university students in urban population all came from the collated and calculated data of the Statistical Yearbook of Chinese Cities (2008–2017).

The economic development gradient index (lnGRADS) selected in this paper refers to a similar treatment method in dealing with the impact of knowledge spillover on economic development [21], and the logarithm of the standard deviation of urban GDP per capita and urban agglomeration GDP per capita GDP was used to represent the difference in urban economic development level within urban agglomerations.

Among them, IC (agglomeration degree) is measured by the degree of urban spatial isolation (IS) to marginality (IR), and the degree of urban isolation (IS) is expressed by the total urban population within a certain spatial range of the city. $P_j$ is the population of the j city. Urban marginality (IR) is measured by the road distance between the city and the core city closest to the urban agglomeration.

$$IC_i = IS/IR = \sum_{j=1}^{n} P_j / IR_{ik} \tag{14}$$

### (2) Basic Model Setting

According to the theoretical model derivation of the influence of various factors on economic growth in Equation (13), the econometric model is set as follows:

$$\ln GDPGR_{i,t} = \beta_0 + \beta_1 \ln TECH_{i,t} + \beta_2 \ln GRADS_{i,t} + \beta_3 IC_{i,t} + \beta_4 \ln FDI_{i,t} + \beta_5 \ln EDU_{i,t} + \beta_6 \ln TE_{i,t} + \mu_i + v_t + \varepsilon_{i,t} \tag{15}$$

where subscript i represents the city and t represents the year. The explained variable lnGDPGR is the urban economic growth rate, and the main explanatory variable is lnTECH, which is the logarithmic

value of the proportion of urban science and technology expenditure to fiscal expenditure. Innovation is mainly concentrated in cities. The increase in the proportion of science and technology expenditure is very important for sustained economic development and transformation. It is also an important indicator for measuring the internal mechanism of urban growth [22]. lnGRADS is the logarithm of the economic development gradient value within the urban agglomeration, and the processing method is as described above.

In order to ensure the reliability of regression results, this paper also added a series of control variables. The IC indicator is the degree of urban agglomeration, and the calculation method is as described above. The level of foreign direct investment (lnFDI) is expressed by the logarithm of the actual amount of foreign capital used in urban areas in the proportion of GDP after the adjustment of the exchange rate of the year. The entry of FDI will affect the size of the city, and at the same time, the factors of urban economic strength will in turn have a strong attraction to FDI [23]. Urban human capital level (lnEDU) is expressed by the logarithm of the number of college students in the urban area. Human capital is an important indicator to measure the cultural quality of the labor force. The higher the concentration degree of human capital, the better the spillover effect of knowledge will be, and the more obvious it will drive and radiate economic development [24]. The industrial structure (lnTE) is expressed by the end of the urban tertiary industry output value in the proportion of GDP. In the face of economic downturn, it is necessary to optimize and upgrade the industrial structure to promote regional economic growth and employment. Economic development transformation $\mu_i$ is an individual dummy variable that represents individual characteristics of micro-observation such as climate, resources, etc. $\gamma_i$ is the time dummy variable, representing the macro-economic impact not observed in time, which was 1 for year t, and 0 for the other years. $\varepsilon_{i,t}$ is the random perturbation term.

Economic development is not only affected by the current economy, but also by historical factors. Therefore, it is necessary to further study economic development from a dynamic perspective. Traditional econometric estimation methods have limitations on their own, and there are sometimes biased and inconsistent estimations. As the lag term and fixed effect of the dependent variable exist simultaneously, to obtain reliable estimations, part of the parameter estimator must meet certain assumptions such as the model of random error term is a normal or a known distribution, so it was necessary to use the generalized method of moment in this paper. The GMM method could allow the heteroscedasticity and sequence correlation of random error terms without the need for accurate distribution information of random error terms, and could obtain accurate model parameter estimation with the lag term as the explanatory variable [25,26]. In order to further explore the influence of previous growth on the current period, differential GMM and system GMM were used for further regression analysis. The dynamic panel regression model is as follows:

$$\ln GDPGR_{i,t} = \beta_0 + \beta_1 \ln TECH_{i,t} + \beta_2 \ln GRADS_{i,t} + \beta_3 IC_{i,t} + \beta_4 \ln FDI_{i,t} + \beta_5 \ln EDU_{i,t} + \beta_6 \ln TE_{i,t} + \beta_7 \ln GDPGR_{i,t-1} + \mu_i + v_t + \varepsilon_{i,t} \tag{16}$$

*(3) Basic Regression Result*

In order to examine the impact of technological level and economic development gradient on the economic growth of urban agglomerations, we conducted a regression test on equation (16), the results are shown in Table 1.

**Table 1.** Basic regression results.

|  | (1) | (2) | (3) | (4) |
|---|---|---|---|---|
|  | **OLS** | **FE** | **DGMM** | **SYSGMM** |
| lnTECH | 0.00273 | 0.00883 | 0.00207 *** | 0.00208 *** |
|  | (0.00040) | (0.00045) | (0.00055) | (0.00048) |
| lnGRADS | −0.436 *** | −1.006 *** | −0.904 *** | −0.981 *** |
|  | (0.13) | (0.12) | (0.152) | (0.153) |
| IC | −0.0483 | −2.261 *** | −1.791 * | −1.667 ** |
|  | (0.086) | (0.366) | (0.814) | (0.640) |
| lnFDI | 0.382 ** | 0.192 | 0.559 *** | 0.53 *** |
|  | (0.140) | (0.125) | (0.145) | (0.131) |
| lnSTUD | −0.000525 | −0.00541 | −0.00185 | −0.000775 |
|  | (0.00043) | (0.00085) | (0.00123) | (0.00115) |
| lnIND | −2.945 *** | −13.90 *** | −12.13 *** | −12.62 *** |
|  | (0.773) | (0.812) | (1.094) | (1.028) |
| L.GDPGR |  |  | 0.426 *** | 0.374 *** |
|  |  |  | (0.0255) | (0.0231) |
| Sample Size | 2020 | 2020 | 1616 | 1818 |
| $R^2$ | 0.0493 | 0.0562 |  |  |
| SARGAN |  |  | 127.857 | 132.0118 |
| P-Value |  |  | 0.0000 | 0.0000 |
| AR (1) |  |  | 0.0000 | 0.0000 |
| AR (2) |  |  | 0.0539 | 0.065 |

Note: *, ** and *** represent the significance levels of 10%, 5%, and 1%, respectively. The values in brackets are standard errors, and all regressions were controlled for time dummy variables. First-order and second-order sequence correlation tests of first-order and second-order autocorrelation (AR (1) and AR (2)) were performed and G values were reported during GMM estimation; overidentification test (SARGAN) examined the validity of the instrument variables and reported the corresponding p values. Data source: organized by this paper.

We included the first-order lagged variable of the explained variable into the regression, and passed the "no autocorrelation in the perturbation term" test at the significance level of 1%. After the overidentification test, it was found that the differential GMM did not have the problem of weak instrumental variables, so the regression results of the two GMM were also reliable. The column (1) of Table 1 reports the ordinary least square regression results under the clustering standard error, which can avoid the heteroscedasticity problems in regression. The fixed-effect model (FEM) assumes that all included studies have a common true effect size, and from time and the individual point of view, the interpretation of the panel data regression model variables to explain the marginal impact is the same. Therefore, we report the regression results of fixed effects in column (2) to reduce the excessive impact of heterogeneity on regression. Columns (3) and (4) of Table 1 report the results of the differential GMM and system GMM, respectively. The lnTECH coefficient was significantly positive in both the OLS regression and GMM regression, indicating that the investment and application of technology will significantly promote the growth. Regarding the symbol of lnGRADS, the coefficient was significantly negative at the level of 1%. Since lnGRADS is an inverse indicator, the narrowing of the economic development gap between cities within the urban agglomeration will promote growth. In the regression of lnFDI, the coefficient was also significantly positive. In the OLS and dynamic GMM regressions, the saliency tests of 5% and 1% were passed respectively, indicating that FDI plays a significant role in promoting urban growth. The results also verified that the contribution of FDI to the economy was positive when studying the contribution of FDI to GDP, and it is more obvious in coastal areas with a good economic base [27]. In addition, the IC coefficient was negative and passed the hypothesis test at the level of 10%, indicating that for all urban agglomerations, agglomeration does not promote the growth of all urban agglomerations, and beyond a certain space, the agglomeration is not conducive to urban growth, but will bring a series of "urban diseases" such as environmental damage, traffic congestion, medical deficiencies, and education injustice. The coefficient of human capital was negative, but not significant, indicating that most cities still failed to give full play to the role of human

capital in economic growth. The improvement of human capital plays an important role in promoting future urban economic growth. The coefficient of industrial structure variable was negative at the level of 1%. A lot of research and experience have proven that the evolution of industrial structure is one of the important factors of economic development [28], however, economic growth and industrial structure have a certain causal relationship and restrict each other. Upgrading the industrial structure will have obvious characteristics of periodic impact on economic growth. Since the development of urbanization, the largest effect on urban economic growth is the development of the second and the third industry, which accounts particularly for the largest proportion of the second industry. The result of model analysis is consistent with the fluctuation of the economic growth trend of economic growth and secondary industry obtained in the previous article and is consistent with the actual situation of China's urban economic development. The economy has been mainly driven by the development of industry. Agriculture has been in a steady development state, laying a foundation for the development of the second and third industries. The third industry, based on service, has been growing slowly for a long time.

*(4) Study on the Mechanism of Agglomeration to Promote Growth*

Agglomeration and diffusion are two stages that urban agglomerations must undergo during the growth process. When agglomeration reaches a certain level, most of the cities need to spread to surrounding cities after their environmental carrying capacity and volume reach a certain level, and even have trans-regional diffusion and transfer. In order to further explore the relationship between urban clusters and growth, the interaction between city scale (Scale) and agglomeration degree (IC) was introduced according to Equation (4) to verify the dispersion effect of urban agglomerations. The interaction term model to test the dispersion effect of urban agglomeration is:

$$\ln GDPGR_{i,t} = \beta_0 + \beta_1 \ln TECH_{i,t} + \beta_2 \ln GRADS_{i,t} + \beta_3 IC_{i,t} + \beta_4 \ln (Scale \cdot IC)_{i,t} + \beta_5 Scale_{i,t} + \beta_6 \ln FDI_{i,t} + \beta_7 \ln EDU + \beta_8 \ln TE_{i,t} + \mu_i + \nu_t + \varepsilon_{i,t} \tag{17}$$

According to similar studies [29], urban scale not only has an important impact on urban agglomerations, but also has an important relationship with urban economic growth. Therefore, the urban scale variable (Scale)is a good explanation for the impact of agglomeration degree on the size of the economy. The important driving force for the development of urban agglomeration is the efforts made by large cities to avoid their own diseconomies of scale and overcome the "urban disease" [30]. Large cities may spread out under the pressure of uneconomic scale, which will cause the urban agglomeration to produce a dispersion effect to alleviate diseconomy of scale. Therefore, the interactive terms of city scale (Scale) and clustering degree (IC) were introduced as $\frac{\partial[\partial(\ln GDPGR)/\partial(Scale)]}{\partial(IC)}$. The improvement of the degree of urban agglomerations can alleviate the negative externalities caused by the expansion of the city scale, which eases the pressure through the dispersion effect. The regression results are shown in Table 2.

In this paper, the dispersion effect in different city scales was further studied. Columns (1) and (2) in Table 2 report the overall dispersion effect of urban agglomerations. The results showed that the coefficient of interaction between the urban scale and agglomeration was negative overall, and a 1% hypothesis test passed under the fixed effect regression, which indicates that it did not produce a good dispersion effect with the urban agglomeration to alleviate the non-scale economy. This paper then conducted a further regression analysis based on the average urban population as a criterion for distinguishing large, medium, and small cities (According to the *Notice on Adjusting the Dividing Standards of Urban Size* promulgated in 2014 for the big cities, sub-sample regression was carried out according to the urban population. From 2007 to 2016, the average urban area with a total population of more than one million is a big city, while the following are small cities). Columns (3) and (4) report the results of the dispersion effect of large cities, and columns (5) and (6) report the dispersion effect of small cities. The results show that the agglomeration degree of big cities has an obvious influence on

the dispersion effect, and that small cities did not show a significant impact. It also further proves that agglomeration promotes the dispersion effect of large cities to alleviate the diseconomies of scale.

**Table 2.** Test results of urban agglomeration dispersion effect.

|  | (1) | (2) | (3) | (4) | (5) | (6) |
|---|---|---|---|---|---|---|
|  | **OLS** | **FE** | **OLS** | **FE** | **OLS** | **FE** |
| lnTECH | −0.000723 | 0.000699 | 0.316 | −0.341 | −0.000849 | 0.00135 |
|  | (0.00048) | (0.00046) | (0.324) | (0.255) | (0.00122) | (0.00121) |
| lnGRADS | −0.444 *** | −0.865 *** | −0.526 ** | −0.552 *** | −0.354 * | −1.153 *** |
|  | (0.124) | (0.117) | (0.169) | (0.135) | (0.165) | (0.212) |
| IC | 0.169 | 0.4 | −0.0927 | 1.216 | 2.611** | −11.87 |
|  | (0.190) | (0.587) | (0.193) | (0.662) | (0.989) | (0.390) |
| SCALE | 0.00168 | 0.00254 | −0.00017 | −0.0014 | 0.00395 | 0.226 |
|  | (0.0010) | (0.0044) | (0.0011) | (0.0044) | (0.018) | (0.170) |
| lnSCALE·IC | −0.319 | −2.383 *** | 0.229 | 4.237 *** | −1.476 * | −7.574 |
|  | (0.175) | (0.358) | (0.328) | (0.706) | (0.581) | (4.026) |
| lnFDI | −0.132 | −0.603 *** | −0.339 * | −0.642 *** | −0.0377 | −0.634 *** |
|  | (0.097) | (0.088) | (0.13) | (0.10) | (0.146) | (0.149) |
| lnEDU | −4.51E-06 | −0.00416 *** | 0.532 ** | −0.351 | −0.00166 | −0.00512 ** |
|  | (0.0004) | (0.0008) | (0.160) | (0.244) | (0.0011) | (0.0019) |
| lnIND | −3.044 *** | −12.71 *** | −2.760 ** | −14.91 *** | −4.173 *** | −11.90 *** |
|  | (0.722) | (0.798) | (1.031) | (1.045) | (0.979) | (1.294) |
| Sample Size | 2020 | 2020 | 1078 | 1078 | 942 | 942 |
| Time Fixed Effect |  | Control |  | Control |  | Control |
| Individual Fixed Effect |  | Control |  | Control |  | Control |
| R | 0.053 | 0.027 | 0.0718 | 0.0198 | 0.0807 | 0.0105 |

Note: *, ** and *** represent the significance levels of 10%, 5%, and 1%, respectively. The values in brackets are standard errors, and all regressions were controlled for time and individual dummy variables. Data source: organized by this paper.

*(5) Growth Effect Test of Urban Agglomeration in the Eastern, Central and Western Regions*

With the implementation of major regional strategies such as the Great Western Development Strategy, the Revitalization in the Northeast, the Rise of Central Region, and East Takes the Lead in Development, regional coordination has been further enhanced. Due to the high urban density and economic development level in the eastern region, the overall strength is relatively sufficient. Although the central region is inland, the overall development speed and quality are better than those in the western region, however, with the promotion of the Great Western Development Strategy and the Belt and Road, from the perspective of economies of scale and urban agglomerations, the western region will rely on the developed transportation network to become a promising economic development frontier. Referring to the division of urban agglomerations [31], 14 cities were divided into northeastern urban agglomerations, eastern urban agglomerations, northwestern urban agglomerations, southwestern urban agglomerations and the central urban agglomerations (Northeast urban agglomeration mainly include the urban agglomeration of the middle and south Liaoning and the Harbin–Changchun urban agglomeration. The eastern urban agglomeration mainly includes the Beijing–Tianjin–Hebei urban agglomeration, the Yangtze River Delta urban agglomeration, the Pearl River Delta urban agglomeration, the Shandong Peninsula urban agglomeration and the urban agglomeration on the west coast of the Taiwan Straits. The central urban agglomeration mainly includes the Central Henan urban agglomeration, the urban agglomeration in the middle reaches of the Yangtze River, and the Beibu Gulf urban agglomeration. The western urban agglomeration mainly includes the Guanzhong Plain urban agglomeration, the Chengdu–Chongqing urban agglomeration, the Hohhot–Baotou–Erdos–Yulin urban agglomeration and the Lanzhou–Xining urban agglomeration) according to scale system and spatial distribution, and the growth differences of urban agglomerations were further studied. Table 3 reports the regression results.

**Table 3.** Regression results of urban agglomeration in the eastern, central and western regions.

| | Eastern | | Northeast | | Central | | Northwest | | Southwest | |
|---|---|---|---|---|---|---|---|---|---|---|
| | **Differential GMM** | **System GMM** | **Differential GMM** | **System GMM** | **Differential GMM** | **System GMM** | **Differential GMM** | **System GMM** | **Differential GMM** | **System GMM** |
| L.GDPGR | 0.407 *** | 0.393 *** | 0.452 *** | 0.363 *** | 0.155 *** | 0.181 *** | 0.519 *** | 0.42 *** | 0.0618 | 0.412 |
| | (0.022) | (0.014) | (0.054) | (0.049) | (0.016) | (0.013) | (0.127) | (0.092) | (0.226) | (0.217) |
| lnTECH | 0.00554 *** | 0.0063 *** | 3.511 | 0.761 | −0.698 *** | −0.837 *** | −0.626 | −0.788 | −3.795* | −1.537 |
| | (0.001) | (0.001) | (1.870) | (1.486) | (0.168) | (0.174) | (2.504) | (1.521) | (1.566) | (1.188) |
| lnGRADS | −0.636 *** | −0.725 *** | −1.762 * | −1.820 *** | −0.508 ** | −0.259 ** | −2.968 * | −3.246 * | −1.960 *** | −1.776 *** |
| | (0.140) | (0.108) | (0.719) | (0.408) | (0.155) | (0.094) | (1.298) | (1.606) | (0.382) | (0.399) |
| IC | 1.688 * | 0.743 * | −20.06 ** | −7.327 ** | −1.120 *** | −0.17 | 38.36 | 1.487 | 2.1 | 0.113 |
| | (0.707) | (0.332) | (6.169) | (2.433) | (0.242) | (0.122) | (42.090) | (4.422) | (4.253) | (1.421) |
| lnFDI | −0.398 *** | −0.431 *** | −1.412 *** | −0.56 | −0.715 *** | −1.269 *** | −0.256 | −0.213 | −0.28 | −0.483 |
| | (0.110) | (0.084) | (0.364) | (0.317) | (0.154) | (0.106) | (0.643) | (0.630) | (0.385) | (0.439) |
| lnSTUD | 0.00416 | 0.00674 *** | 1.468 ** | 1.778 * | −3.019 *** | −1.959 *** | −0.0212 | −0.0238 | −7.627 | −1.65 |
| | (0.002) | (0.002) | (0.487) | (0.715) | (0.443) | (0.268) | (0.020) | (0.018) | (5.203) | (3.111) |
| lnIND | 11.11 *** | 9.252 *** | −6.865 | −17.98 ** | −14.93 *** | −11.26 *** | −15.16 *** | −11.38 ** | −5.895 | −3.667 |
| | (1.247) | (1.059) | (5.486) | (5.521) | (0.886) | (0.756) | (4.607) | (4.408) | (3.491) | (4.724) |
| Sample Size | 728 | 819 | 152 | 171 | 455 | 512 | 152 | 171 | 128 | 144 |
| SARGAN Test | 74.70 | 77.53 | 14.785 | 15.170 | 46.024 | 53.280 | 13.198 | 16.439 | 11.600 | 14.514 |
| *p* Value | 0.0001 | 0.0010 | 0.9989 | 1.0000 | 0.0006 | 0.1354 | 0.9700 | 0.9900 | 0.9900 | 1.0000 |
| AR (1) | 0.0000 | 0.0000 | 0.1248 | 0.1404 | 0.0001 | 0.0001 | 0.0008 | 0.0033 | 0.0000 | 0.0137 |
| AR (2) | 0.5762 | 0.6081 | 0.5757 | 0.4679 | 0.2576 | 0.2518 | 0.0630 | 0.1194 | 0.0016 | 0.2355 |

Note: *, ** and *** represent the significance levels of 10%, 5% and 1%, respectively. The values in brackets are standard errors, and all regressions were controlled for time dummy variables. First-order and second-order sequence correlation tests of AR (1) and AR (2) were performed and G values were reported during GMM estimation; SARGAN examined the validity of the instrument variables and reported the corresponding p values. Data source: organized by this paper.

From the results of differential and systematic GMM regression, the sign of the technical coefficient of urban agglomerations in the eastern and northeastern regions was significantly positive, and a 1% hypothesis test was passed, indicating that technical expenditure and R&D had significant growth effects in urban agglomerations in the two regions. The coefficient of lnGRADS was also in line with the expected hypothesis, the economic development gap between cities was small, and the regional synergy was stronger. The agglomeration coefficient of the eastern region was positive at the significance level of 10%, indicating that the cities in the eastern region had good agglomeration degree, which promotes the growth of urban agglomerations, and the human capital and industrial structure coefficients were also significantly positive at the level of 1%. Cities in the eastern region have turned human capital and industrial advantages into growth drivers, with a higher overall regional growth level. The coefficient of FDI was negative, indicating that the current growth momentum of urban agglomerations in the eastern region mainly came from endogenous growth and regional synergy, and the promotion of foreign capital was significantly weakened. Due to the differences in the number of cities, industrial structure, and urban size of urban agglomerations in the northeast and central regions, with the exception of the two main explanatory variables that conformed to the expected hypothesis, all the other variables showed different influences. The agglomeration degree coefficient, industrial structure coefficient, and FDI were all significantly negative, which had a negative impact on growth. The human capital coefficient was significantly positive in the northeast region, while it was negative in the central region and both passed the 1% hypothesis test, indicating that human capital has been significantly promoted in the northeast region, while the technical variables and human capital variables in the central region were negative, indicating that the cities in the central region have not focused on improving technology, and the promotion of population advantage was not obvious. In the future, the development of urban agglomerations in the central region should further improve the overall quality of the population while continuing to promote the growth of technology and further reduce the gap with the eastern region. The northeast region needs to further strengthen the agglomeration of the cities, through policy orientation and cooperation, and promote growth through policy orientation, cooperation and coordination. The technical, human capital, FDI and industrial structure variables of the urban agglomerations in the northwestern region and the southwestern urban agglomerations were all negative, but it is worth noting that the agglomeration coefficient was positive in both regions, indicating that the promotion of the Western Development Strategy and the implementation of the strategy of the Belt and Road have promoted the growth of the city's agglomeration, and further promote the future economic growth under the support of the two regional transportation networks and unique location advantages.

*(6) Test on Growth Effect of Urban Agglomerations with Different Number of Core Cities*

The development of urban agglomeration cannot be separated from the radiation and driving effect of core cities. Core cities are not only the highest level of urban system planning and setting in China, but also an indispensable and important part of the strategy of developing urban agglomerations. Core cities need to play a leading role in urban agglomeration to promote sustainable growth, regional innovation, and participation in international competition. What type of urban agglomeration has a sustainable competitive advantage? In order to further explore the future growth pattern of urban agglomerations, according to the latest urban agglomeration development framework, 14 national-level urban agglomerations were divided into single-core radiation type urban agglomerations (Single-core radiation type urban agglomerations include the Central Henan urban agglomeration, the Guanzhong Plain urban agglomeration, the Beibu Gulf urban agglomeration, the Lanzhou–Xining urban agglomeration, and the Hohhot–Baotou–Erdos–Yulin urban agglomeration) (led by a central city, and the urban agglomeration scale and spatial agglomeration level are low), dual-core catch-up urban agglomerations (Dual-core catch-up urban agglomerations include the Beijing–Tianjin–Hebei urban agglomeration, the Chengdu–Chongqing urban agglomeration, the Harbin–Changchun urban agglomeration, the urban agglomeration of the middle and south Liaoning, and the Shandong

peninsula urban agglomeration) (with two core cities jointly driving the development of urban agglomerations, with an outward-oriented economy and high level of integration), multi-core open urban agglomerations (Multi-core open urban agglomerations include the Pearl River Delta urban agglomeration, the Yangtze River Delta urban agglomeration, the urban agglomeration on the west coast of the Taiwan Straits, and the urban agglomeration in the middle reaches of the Yangtze River. Note: *The National New Urbanization Plan* and the *2015 Government Work Report* set the core cities of each urban agglomeration) (the location advantage is outstanding, there are more than two core cities, mature, and highly open), according to the number of core cities and development models. Model (3) was further verified by sub-samples, and the dynamic GMM regression results are shown in Table 4.

**Table 4.** Test results of the growth effect of urban agglomerations with different core cities.

| | Single-core Radiation Type | | Dual-core Catch-up Type | | Multi-core Open Type | |
|---|---|---|---|---|---|---|
| | **Differential GMM** | **System GMM** | **Differential GMM** | **System GMM** | **Differential GMM** | **System GMM** |
| L.GDPGR | 0.332 *** | 0.281 *** | 0.515 *** | 0.456 *** | 0.453 *** | 0.415 *** |
| | (0.022) | (0.017) | (0.0172) | (0.0124) | (0.016) | (0.011) |
| lnTech | 0.00388 *** | 0.00478 *** | 0.253 | 0.0497 | 0.002 | 0.00504 *** |
| | (0.0012) | (0.0009) | (0.216) | (0.121) | (0.0011) | (0.0006) |
| lnGrads | −0.620 *** | −0.818 *** | −1.266 *** | −1.734 *** | −0.192 * | −0.310 *** |
| | (0.185) | (0.151) | (0.112) | (0.0765) | (0.089) | (0.080) |
| IC | −2.709 *** | −0.841 *** | −2.988 *** | −2.490 *** | 0.841 | 0.739 ** |
| | (0.644) | (0.224) | (0.745) | (0.514) | (0.552) | (0.229) |
| lnFDI | 0.444 *** | 0.397 ** | 0.075 | 0.011 | 1.259 *** | 1.734 *** |
| | (0.128) | (0.127) | (0.0972) | (0.0715) | (0.149) | (0.095) |
| lnEdu | −0.00365 | −0.00913 *** | 0.00752 | 0.0125 *** | 1.431 ** | 0.0194 |
| | (0.0030) | (0.0023) | (0.0064) | (0.0036) | (0.472) | (0.124) |
| lnInd | −13.69 *** | −12.83 *** | −8.903 *** | −10.27 *** | −13.60 *** | −9.613 *** |
| | (0.924) | (0.920) | (1.076) | (0.791) | (0.912) | (0.590) |
| Constant | 68.91 *** | 68.20 *** | 54.12 *** | 60.81 *** | 65.67 *** | 51.51 *** |
| | (2.881) | (2.897) | (4.5340) | (3.3880) | (3.567) | (2.282) |
| Sample Size | 672 | 756 | 448 | 504 | 384 | 432 |
| SARGAN Test | 61.915 | 66.908 | 52.894 | 52.979 | 55.453 | 57.318 |
| Z Value | 0.0033 | 0.0112 | 0.0267 | 0.1416 | 0.0153 | 0.0708 |
| AR (1) | 0.0000 | 0.0000 | 0.0167 | 0.0178 | 0.0000 | 0.0000 |
| AR (2) | 0.3725 | 0.4540 | 0.336 | 0.362 | 0.1832 | 0.2107 |

Note: *, ** and *** represent significance levels of 10%, 5%, and 1%, respectively. The values in brackets are standard errors, and all regressions were controlled for time dummy variables. First-order and second-order sequence correlation tests of AR (1) and AR (2) were performed and G values were reported during GMM estimation; SARGAN examined the validity of the instrument variables and reported the corresponding p values. Data source: organized by this paper.

From the regression results in Table 4, only the coefficient of technological level, FDI, and economic development gradient difference of single-core radiation urban agglomeration passed the hypothesis test of 1%. That is, technology has promoted the growth of single-core urban agglomerations. The entry of FDI has also created a driving force for the growth of urban agglomerations, and the differences in economic development between cities need to continue to shrink in order to effectively promote growth. The agglomeration, human capital, and industrial structure coefficients all showed negative at the significance level of 1%, indicating that the single-core urban agglomerations were not clustered, the human capital advantage was not well played, and the industrial structure was slow to adjust, which makes the growth momentum insufficient. Since there was only one core city in a single-core urban agglomeration, it was difficult to achieve coordination and unification of production, life, and ecology in the whole region. Due to the influence of spatial and geographical factors, it may be difficult for edge cities to obtain the transfer of factors in core cities, and as a result, the divergence of urban development level in urban agglomerations is more and more prominent, and the growth of peripheral

cities is slow. There was only one core city like the Central Henan urban agglomeration and the Guanzhong Plain urban agglomeration. These core cities have limited resources and the economic development level was in the second-line status. It is difficult to spread the advantageous resources to the surrounding areas. The gap between these core cities and other cities in the urban agglomerations was obvious, and had a more obvious siphon effect, which may bring the superior resources of the surrounding cities to the core cities, resulting in the growing difference between the surrounding areas and the core cities. The overall growth of the urban agglomerations was relatively slow.

Dual-core catch-up urban agglomerations have two core cities. From the regression results, the technical level coefficient was positive, but not significant, indicating that the driving force of technology to promote growth needs to continue to increase. The economic gradient coefficient was in line with expectations, that is, the economic development gap between cities needs to be further narrowed to create sustained momentum for growth. The FDI and human capital coefficient also passed the hypothesis test at the significance level of 10%, indicating that FDI and human capital have a significant boost to the growth of the dual-core catch-up urban agglomerations. However, the clustering and industrial structure coefficients were both negative, indicating that in the process of urban agglomeration, the industrial structure upgrading has limited effect on the growth of urban agglomerations. The two core cities are often the two poles within the urban agglomeration, which leads to differences in administrative division and regional coordination, resulting in certain adverse effects on growth. For example, the Beijing–Tianjin–Hebei urban agglomeration has two core cities, Beijing and Tianjin. There are two core cities in Harbin and Changchun in the Harbin–Changchun urban agglomeration. Because of the different administrative territories, the dual-core cities of different urban agglomerations have great differences in regional policy formulation and territorial jurisdiction. Local protection barriers are still obvious, and the radiation-driven effects of core cities are difficult to fully exert, and it is difficult for these urban agglomerations to achieve effective growth, which is consistent with the previous findings on administrative barriers.

Except that the industrial structure coefficient of multi-core open urban agglomeration was negative, the other explanatory variables were positive. The variables of technology and cluster degree divided passed the significance test of 5% and 1%, indicating that the investment of technology and urban agglomeration significantly promoted the economic growth of multi-core open urban agglomeration. At the significance level of 1%, FDI positively promoted the growth effect of multi-core open urban agglomeration. Meanwhile, the coefficient of human capital also passed the hypothesis test of 5%, which proves that continuous input of human capital can promote the growth mode of multi-core open urban agglomeration to form a virtuous circle in the development process. The industrial structure coefficient was negative. Therefore, the multi-core open urban agglomerations need to consider the urban industrial structure optimization and transformation and upgrading in the future growth. For example, as a multi-core urban agglomeration, the Pearl River Delta urban agglomeration has core cities including Guangzhou, Shenzhen, Zhuhai, and other central cities. The Pearl River Delta is dominated by Guangdong Province, and there is no administrative barrier like that between the cities of the Beijing–Tianjin–Hebei urban agglomeration. Therefore, the radiation of the core city was more effective. The Pearl River Delta urban agglomeration is still dominated by manufacturing industries such as the ceramics industry in Foshan city and clothing industry in Humen city. The manufacturing industry's pulling effect on the Pearl River Delta economy is still outstanding. Next, how to further promote industrial transformation and upgrading is the top priority of urban development in the Pearl River Delta. With the promotion of the Guangdong, Hong Kong, and Macau Bay Area strategy, Guangzhou, Shenzhen, and Hong Kong will be the core cities, which will further expand the space for regional cooperation. However, how to effectively promote the coordinated development among regions may also be an important challenge to the Guangdong–Hong Kong–Macao Greater Bay Area.

## 4. Robust Analysis

To verify the reliability and stability of the previous conclusions, the model was further tested for robustness from two perspectives. The structural form of the model index has a certain influence on the model result. Here, the core explanatory variable, the index of the gradient value of technological and economic development, was transformed into the construction method. As for technical indicators, we referred to the logarithm of the science and technology expenditures of the municipal districts used by a similar study on the relationship between technological progress and economic contribution [32], namely lnTECH*.

Regarding the economic development gradient indicator, the GDP per capita determines the social structure, population quality, and quality of life of the region to a large extent. Therefore, we drew on the logarithm of the absolute value of the difference between the urban per capita GDP of each city and the average per capita GDP of urban agglomerations used by relative researches [33], as shown in Equation (18). The larger the value of lnGRADS*, the greater the difference in economic development between cities.

$$\ln GRADS_{i,t}{}^* = \ln \left| \frac{GDP_{i,t}}{Popu_{i,t}} - \frac{\sum \frac{GDP_{i,t}}{Popu_{i,t}}}{N} \right| \tag{18}$$

On the other hand, to verify whether the GMM estimation result was robust, mixed OLS regression, fixed effect regression, and random effect results are presented here as a comparison. The robust test results of the growth effect of the whole sample urban agglomeration are shown in Table 5.

**Table 5.** Results of the robust test of the growth effect of urban agglomeration.

|  | (1) | (2) | (3) | (4) | (5) |
|---|---|---|---|---|---|
|  | OLS | FE | RE | DGMM | SYSGMM |
| lnTECH * | 0.000701 | 0.000789 | 0.00079 * | 0.00222 *** | 0.002 *** |
|  | (0.00048) | (0.00047) | (0.00038) | (0.00056) | (0.00048) |
| lnGRADS* | −0.357 ** | −0.716 *** | −0.491 *** | −0.672 *** | −0.781 *** |
|  | (0.118) | (0.109) | (0.0931) | (0.139) | (0.139) |
| IC | 0.0974 | −1.976 *** | 0.126 | −1.788 * | −1.548 ** |
|  | (0.0978) | (0.366) | (0.126) | (0.745) | (0.573) |
| lnFDI | −0.154 | −0.67 *** | −0.363 *** | −0.504 *** | −0.591 *** |
|  | (0.0917) | (0.0889) | (0.0754) | (0.126) | (0.122) |
| lnSTUD | −0.000261 | −0.00503 *** | 0.000111 | −0.00161 | −0.00064 |
|  | (0.0004) | (0.0008) | (0.0004) | (0.0011) | (0.0011) |
| lnIND | −2.799 *** | −13.47 *** | −5.038 *** | −12.65 *** | −13.26 *** |
|  | (0.720) | (0.808) | (0.543) | (1.167) | (1.076) |
| L.GDPGR |  |  |  | 0.423 *** | 0.376 *** |
|  |  |  |  | (0.0274) | (0.0244) |
| Constant | 27.75 *** | 81.35 *** | 39.46 *** | 67.01 *** | 70.91 *** |
|  | (2.525) | (3.177) | (2.235) | (5.261) | (4.831) |
| Sample Size | 2020 | 2020 | 2020 | 1616 | 1818 |
| R² | 0.0457 | 0.0267 | 0.0443 |  |  |
| SARGAN Test |  |  |  | 139.7924 | 144.2376 |
| Z Value |  |  |  | 0.0000 | 0.0000 |
| AR (1) |  |  |  | 0.0000 | 0.0000 |
| AR (2) |  |  |  | 0.0935 | 0.1268 |

Note: *, ** and *** represent significance levels of 10%, 5%, and 1%, respectively. The values in brackets are standard errors, and all regressions were controlled for time dummy variables. First-order and second-order sequence correlation tests of AR (1) and AR (2) were performed and G values were reported during GMM estimation; SARGAN examined the validity of the instrument variables and reported the corresponding p values. Data source: organized by this paper.

From the results of the robust test, it can be found that after replacing the core explanatory variables, the coefficient size and symbol of the control variables remained basically the same, and the coefficient symbols of the technical and economic gradients were completely consistent, and only the numerical magnitude and the degree of significance were slightly different. Additionally, from the results of the fixed effect, random effect, and mixed effect regression, although the $R^2$ was not large, the regression coefficients of each equation were mostly significant, the coefficient symbols were more consistent, and the overall model were more robust. Therefore, it can be considered that the results of the previous calculations on the growth effect of urban agglomerations were accurate and reliable.

## 5. Further Discussions

As analyzed above, the growth effect of single-core and dual-core urban agglomerations lagged behind that of multi-core urban agglomerations. Is this because of the poor accessibility between the central cities and the edge cities that the core cities have limited radiation driving force to the edge cities? In order to further explore the internal mechanism of the influence of the central accessibility on the growth of urban agglomerations, this paper conducted a regression on the influence of the central urban accessibility on the growth of urban agglomeration through the threshold panel regression.

Referring to the relative work of core cities and surrounding cities [34], this paper constructed a threshold model based on the average geographical distance of the edge cities in each urban agglomeration from the central cities, as shown in Equation (19), and through the threshold model estimation and significance and authenticity test, as shown in Table 6.

$$
\begin{aligned}
GDPGR_{i,t} = \theta_0 + \theta_1 \ln TECH_{i,t} + \theta_2 \ln Grads_{i,t} + \theta_3 \ln D_{i,t} G(D \leq \gamma_1) + \theta_4 D_{i,t} G(\gamma_1 < D \leq \gamma_2) \\
+ \theta_5 D_{i,t} G(D \geq \gamma_2) + \theta_6 IC_{i,t} + \theta_7 \ln FDI_{i,t} + \theta_8 \ln EDU_{i,t} + \theta_9 \ln TE_{i,t} + e_{i,t}
\end{aligned}
\tag{19}
$$

where $GDPGR_{i,t}$ refers to the urban economic growth rate; lnGRADS is the core explanatory variable of Equation (19); and other explanatory variables are consistent with the above, and will not be described again. Both $\gamma_1$ and $\gamma_2$ are threshold values, and $D_{i,t}$ is the average road distance from the edge cities to the central cities within the urban agglomeration. For the dual-core and multi-core urban agglomeration road distance (Note: The distance from the edge cities to the central cities in the urban agglomeration was derived from zuoche.com) determination, the distance from the nearest central city within the urban agglomeration was taken.

**Table 6.** Parameter estimation results of the threshold model.

| Threshold Model | Variables | Estimated Value of Coefficient | Standard Deviation | t Statistical Magnitude | p Value |
|---|---|---|---|---|---|
| | Distance < 116.1 | −0.2944 ** | 0.2271 | −1.2961 | 0.1951 |
| | 116.1 < Distance < 349.55 | −1.2759 *** | 0.3519 | −1.3697 | 0.1710 |
| | Distance > 349.55 | −0.482 ** | 0.1352 | −9.4401 | 0.0000 |
| Growth Effect Threshold Model | lnTECH | 0.0006 ** | 0.0005 | 1.1432 | 0.2531 |
| | IC | −1.9225 *** | 0.3333 | −5.7676 | 0.0000 |
| | lnFDI | −0.6467 *** | 0.1120 | −5.7752 | 0.0000 |
| | lnEDU | −0.0048 ** | 0.0009 | −5.2078 | 0.0000 |
| | lnTE | −13.0233 * | 1.0191 | −12.7795 | 0.0000 |

Note: *, ** and *** respectively indicate that the regression coefficients of major influencing variables passed the significance test at the levels of 10%, 5%, and 1%. Data source: organized by this paper.

Table 6 lists the parameter estimation results of the threshold model. From the regression results of model (19), it can be seen that the two thresholds of geographic distances, 116.1 and 349.55, divided

the geographical distance between the central city and the edge city of 14 urban agglomerations into three scale ranges. Under the geographical distance between different edge cities and central cities, the growth effects of urban agglomerations were significantly different. When other cities in the urban agglomeration were within 116.1 km from the central city, the regression coefficient was negative and passed the 5% significance test, indicating that the accessibility of other cities and central cities within the urban agglomeration within 116.1 km was conducive to a better growth effect, and lnGRADS contributed the most to economic growth. When other cities in the urban agglomeration were 116.1 km–349.55 km away from the central city, the regression coefficient was significantly negative. It can be seen that in this interval, the distance between the surrounding cities and the central cities increased, the accessibility became longer, and the economic gradient within the urban agglomerations was further widened, which is not conducive to a good growth effect. When the second threshold was exceeded, the geographical distance of the urban agglomeration increased the negative impact on the economic gradient of the urban agglomeration, and a 1% hypothesis test was passed, indicating that if the surrounding cities are too far away from the central city, they may cooperate and integrate with the neighboring cities, so that the negative impact of economic gradient on growth will be weakened to some extent [35]. In general, with the extension of the accessibility of central cities, the growth effect of urban agglomeration went through three stages of "gradually weakening–worsening–improving", and the economic gradient of urban agglomeration had an inverted "N" relationship with the accessibility of central cities.

## 6. Conclusions and Suggestions

Based on the panel data of 14 national-level urban agglomerations in China from 2007 to 2016, this paper measured the growth effects of urban agglomerations and the main influencing factors, and the conclusions are very robust. From the perspective of an action mechanism, urban agglomeration not only gathers many cities in geographical space, but also produces good economies of scale effect. Technology is very obvious for the growth of urban agglomerations. The improvement and application of technology can promote the good growth effect of urban agglomerations. The narrowing of the difference in urban economic development within urban agglomerations is conducive to optimizing economic structure and promoting the stable growth of urban agglomerations. The increase of the agglomeration degree can promote the coordinated growth of the city, improve the utilization rate of the factors, promote the rational allocation of resources, and improve the degree of urban clusters, which can be negative externalities caused by the expansion of the city scale, and alleviate the "crowding effect" by generating a dispersion effect to surrounding cities. The utilization of human capital and the optimization and upgrading of industrial structure will have a positive effect on the growth of urban agglomerations. Meanwhile, the study also found that FDI significantly weakened the role of urban growth, and the driving force for urban agglomeration has gradually shifted from exogenous to endogenous. From the perspective of the growth pattern of urban agglomerations, the growth momentum of multi-core urban agglomerations was more significant than that of single-core and dual-core urban agglomerations. Technology, clustering, FDI, and human capital all significantly promoted the growth of urban agglomerations. The growth momentum of urban agglomerations in the eastern region was significantly stronger than that in the central and western regions. Clustering, technology, and human capital contributed more to the growth of urban agglomerations than the other two regions.

For urbanization in China and other developing countries facing problems, policymakers need to, according to the resources endowment and the geographical location of different cities, have a high efficiency, reasonable use of land, so it is essential to establish scientific and rational urban renewal and the framework of guidelines, under the certain condition of existing urban agglomeration center cities [36], utilize the diffusion effects of central cities, optimize the formulation of the corresponding planning and implementation of land intensive utilization, and maximizes the growth of the urban agglomeration effect. Based on the research results of this paper, we propose the following policy suggestions:

First, the strategy of developing urban agglomerations has produced significant growth effects. If the goal is solely to promote urban economic growth, this research suggests that policy should continue to focus on the role of urban agglomeration, which could actively break down regional isolation and regional administrative barriers, strengthen inter-regional synergy and policy sharing, and promote growth through synergies. Second, it is important to continuously construct core cities by developing core cities with higher levels to generate key elements to edge cities, and stimulate regional economic vitality by narrowing the gap between urban economic development, and transfer the current single-core and dual-core model urban agglomeration to multi-core urban agglomeration to produce multi-polar effects. Third, it is necessary to narrow the economic development difference between the eastern, central, and western regions. For the urban agglomerations in the western region, there is no doubt that promoting the implementation of the Western Development Strategy, vigorously developing the economy, and gradually shortening the gap with the development of urban agglomerations in the eastern and central regions through policy guidance, technical support and talent introduction. For the urban agglomerations in the central region, it is vital to promote reforms on the supply-side and upgrade the level of industrial structure, which need the market to give full play to the basic allocation of resources. The urban agglomerations in the eastern region are in need of controlling the scales of cities, and actively promoting the development of industrial integration. It is also necessary to continue to promote the development of high-end and high-efficiency industries to produce sustained economic growth by means of promoting the integration of the Internet, big data, artificial intelligence, and the real economy. Fourth, in conjunction with the implementation of major regional development strategies such as the Belt and Road Initiative, the Hong Kong–Zhuhai–Macao City Circle, and the Yangtze River Economic Belt, and by way of promoting the growth of technology, transportation location, and human capital, giving full play to the radiating role of the core cities could actively and finally achieve coordinated, high-quality, integrated growth in the future.

**Author Contributions:** The conceptualization of this paper and writing—review and editing of the paper was contributed by J.H.; M.G. is the main writer of this paper, and also contributed to the writing and original draft preparation and also did the formal analysis and investigation; Y.S. assisted in the methodology and software and data curation.

**Funding:** This research received no external funding.

**Acknowledgments:** This work was supported by the Science and Technology Strategy Research Beijing Key Laboratory of Urban Green Development (160151101).

**Conflicts of Interest:** The authors declare no conflict of interest.

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
