# Peer review of "Research on the Measurement and Path of Urban Agglomeration Growth Effect†"

_sustainability, doi:10.3390/su11195179_

Round 1

Reviewer 1 Report

In general, this is a very interesting and relevant piece of research. I appreciate the extent to which quite a few different issues and concerns were investigated using a range of econometric approaches. I also appreciate the robustness checks. 

I have noted some instances of over-claiming based on the findings, but generally, the analysis presented here seems to make a strong contribution to understanding how agglomeration dynamics tend to impact urban economic growth. I have also noted areas where the language needs some additional detail and precision. My specific comments are as follows: 

Line 58: This paragraph makes a number of claims without references to support.

Line 69: What is the “third technological revolution”?

Line 81-84: I’m not sure about the universality of the central cities being endogenously formed through agglomeration. The three references are quite well-known, but it is important to be specific about how they support the point. A bit more detail in the description would be helpful.

Line 87: Please be clear about what you mean by “urban growth”. Do you mean economic growth? What are the boundaries of the concept?

Line 102: This paragraph has a number of concepts that need to be clarified. What is agglomeration degree? What is growth momentum? What is growth effect?

The word obvious is used too often. Consider more precise alternatives.

line 179: this paragraph needs to be made more clear. The first sentence appears to be a little circular - “Agglomeration plays a significant role in… agglomerations.”

Line 157: says IC is the agglomeration degree in the model. Line 163 seems to refer to this as clustering. Greater clarity is needed on these concepts.

Line 162: should probably make clear that “technological level” is represented by A in the equation (presumably).

Line 167: what is meant by “more significant the growth effect is”? Does this refer to statistical significance?

Line 165: not very appropriate to refer to positive or negative growth effect as “good or bad”.

Line 171: for Hypothesis 2, it’s not clear which variable(s) represent the “urban economic development gradient within the urban agglomeration”. This hypothesis needs to be clarified in relation to the model.

Line 177: for Hypothesis 3, what is meant by “critical point of agglomeration”? This seems to be answered in the following paragraph, but a little more clarity would be helpful.

Generally, section 2 lacks clarity in its explanations. Technical language is used without introduction for a general reader (the target audience of this journal is likely not expected to have expertise in econometrics), and it is not used with consistency and precision.

Line 226: human capital is represented by the number of college students in the area. Are there not other things for which data exists, like number of people with university degrees, based on census data or other reliable survey data? Presumably the number of active university students will be correlated with the number of universities, which will be correlated with the size and prominence of the city. I understand this is meant to be a control variable, but I’m not sure it doesn’t bring problems of multicollinearity into the equation. Should this be the number of college students per capita?

Line 230: What are the “urban three industries”? Later in the paragraph, it’s not clear what is included in the mu variable or where this data comes from.

Line 236-239: This paragraph doesn’t explain what GMM (generalised method of moments?) is or why this approach would work to account for the influence of previous time periods of growth in the current period. Please explain this.

Line 260: Please include, at least in a footnote, what the FE (fixed effects) regression does and why it reduces the impact fo heterogeneity.

Line 270-273: Firstly, Significance at the 10% level is rather weak. Secondly, based on your stated hypothesis, you expected this variable to behave in a non-linear way. Here you have tested with several versions of a linear model the linear contribution of IC. I don’t think your interpretation of the results is accurate. Or, at least, you have not explained it sufficiently to be convincing. You need to explain why the sign is different between the OLS and the other regressions. If you expect there to be some “critical value” at which the effect of IC starts to change direction, I think you need a different approach to getting evidence from the data. Could you plot the effect? You need additional evidence to make the interpretation that this indicates “urban diseases”.

Line 276-282: You didn’t explain earlier what the industrial structure variable really is, so it’s not clear now that you’re interpreting it. Please explain what you mean by primary, secondary, and tertiary industries.

Line 286-288: This relationship between agglomeration, spreading, and environmental carrying capacity seems to come from the literature somewhere. Please give references.

Line 338: “symbol” should be “sign”

Line 436-449: I appreciate the effort to comment on the ways the findings from this analysis can be applied to policy. However, greater care needs to be taken in the writing to clearly state the actual findings (from the econometrics) and the hypothetical extensions of these findings to applications (such as commenting on the role played by administrative boundaries between the cities in the Beijing agglomeration).

Line 470: Good point that the R2 is not large. This is consistent with the results in Table 1 as well. Please comment on the implications of such a low R2. Even though your coefficients are largely significant, the model appears to explain only a very small percentage of what impacts urban growth. Can you make reliable conclusions for policy based on this?

Line 490: presumably you meant Table 6.

Section 5: very interesting section, which finally gives credence to your “hypothesis 3”. Could you explain how you determined these critical values? You appear to have set them and then run the model. Did you find them by trial and error, or did you follow some process?

Line 544: The paragraph beginning with this line adopts a rather unprofessional tone. From explanation of what influences growth, the tone shifts to “we must” language, such as “we must unswervingly promote…”. This is not appropriate for a technical, professional article, as it editorialises. Please change the wording so it is less absolutist. Rather than “China’s future urbanization development should…”, I suggest something along the lines of “If the goal is solely to promote urban economic growth, this research suggests policy should continue to focus on the role of urban agglomeration…”

Completely subsuming this research within a particular set of policy goals limits the scope and validity of the findings.

Reviewer 2 Report

The paper presents the effect of the Measurement and Path of Urban Agglomeration Growth Effect.

The abstract is really focused o the results. That’s interesting, but not common. I ask you to explicate also the aims and the research methodology of the paper.

The introduction is very interesting, because is strictly connected on China policies. Than you have to explicate the sims of your research and the gap in the literature. Why you research is useful for the scientific community? Could you compare your results and methodology with other papers, referring to it directly in the text?

Paragraph 2 is very hard to understand for not technical people. Could you explain better it?

The maid not clear. Could you add a part describing the methodology used? You insert a part in the introduction, but a paragraph dedicato to it could help the understanding of your research.

It is not clear which calculation procedure do you use for the building stock analysis of the different town. Did you do a cluster analysis for organizing and grouping the data? You can refer to the following paper: Lucchi et al., A Density-Based Spatial Cluster Analysis Supporting The Building Stock Analysis In Historical Towns, Building Simulation, Rome, 2019. The discussion of the data and the results is very clear. Conclusions are clear because they contain the most important finding of your research. Bullets points can help you to improve their legibility.

Reviewer 3 Report

This is a well written paper. However, I would suggest the authors to follow the below mentioned suggestions for further improving the paper.

In the introduction section, please narrate the motivation of this reserach Before the conclusion section, please bring a new section called managerial implications Refer "Manupati, V. K., Ramkumar, M., & Samanta, D. (2018). A multi-criteria decision making approach for the urban renewal in Southern India. Sustainable Cities and Society42, 471-481." in your paper

Round 2

Reviewer 2 Report

-